# OpenReview forum: "Effective Backdoor Mitigation in Vision-Language Models Depends on the Pre-training Objective"
_TMLR — Accepted by TMLR_

### Review · Reviewer_qLae · 2024-07-11

**Summary Of Contributions:**

The paper investigates data poisoning of VLM’s on uncurated image text data. It builds on top of two prior works:

* **BadNet**: An adversary injects a small patch and replaces the caption with an adversarial target label. At test time, the adversary can inject the same patch and predict the desired target label with a high success rate (ASR).
* **CleanCLIP**: The poisoned model can be “cleaned” by finetuning on a combination of SSL + text-image contrastive losses. The resultant models mostly preserve zero-shot accuracy with a low ASR.

The models investigate CleanCLIP but if the self-supervised loss is incorporated in the pretraining stage. The investigation is done on train-from-scratch and pretrained clip models on CC-3M and CC-6M. The consistent observation is that CleanCLIP now has to lose a lot of accuracy to achieve a low ASR rate. The paper further experiments with heavy regularizations, more training data, clustering fine-tuning using to show that this observation persists. Based on their experiments, they recommend to use only standard contrastive losses during training to avoid potential data poisoning

**Audience:**

Yes

**Broader Impact Concerns:**

No broader impact concerns

**Claims And Evidence:**

Yes

**Requested Changes:**

* **Critical**: Throughout, the paper uses "stronger pretraining objectives". This term is pretty vague and can mean a variety of different pretraining objectives. However, the paper investigates only a specific form of self-supervised pretraining. The paper can tone down on its claim and be specific and replace "stronger pretraining objectives" with self-supervised losses throughout the paper.

* **Critical**: Page. 6 has this statement "Even though the models trained with  in order to have better
visualization of the difference in performance of CleanCLIP on the two pre-training objectives, we deliberately
choose models with similar starting accuracies". How much were the higher accuracies? If it's sufficiently high, where do the cleaned models with the stronger accuracies lie in Figure 2? I understand the motivation of the authors but from a practical point of view, it makes sense to just deploy the highest performing model. So this result is also necessary.

* **Critical**: Figure 3 is  confusing because there are two "finetuned". One finetune is with the poisoned data and the second finetune is with the CleanCLIP fine-tuning. I suggest that the authors replace "finetuning" in all legends with just CleanCLIP. For example, "Finetuning with MMCL + l2" can be replaced with "CleanCLIP: MMCL only + l2", "Finetuning with MMCL + SSL + l2" can be replaced with "CleanCLIP + l2" and so on.


* **Strengthen**: The paper can add some intution on why incorporating SSL loss during pretraining has this effect. My understanding is as follows. "CleanCLIP focuses on breaking correlation between by having meaningful unimodal representations during fine-tuning. Since the SSL loss is also applies on the poisoned images, during pretraining, the unimodal representations can also be implicitly associated with the target image."

* **Strengthen**: I assume the paper borrows the augmentation hyperparameters from CleanCLIP, since it is not mentioned anywhere. It can be interesting to see what happens if stronger augmentations are applied during pretraining, since stronger augmentations may also distort the small patch. If the augmentations used is RandAugment, then the strength can be directly controlled.

* **Strengthen**: What happens if the self-supervised loss during pretraining is applied only on clean images? Do the insights still hold? One can imagine a scenario where the contrastive model is trained on uncurated data while practitioners can apply the SSL loss on a small set of curated images (For example, ImageNet)

* **Strengthen**: The experiments can also be strengthened by having one experiment on ViT models.

* What is the precise difference between BadNet and "label-consistent" data poisoning? For "label-consistent" data poisoning, do the authors search for the captions that have "banana" in them in the pretraining dataset and add gaussian noise to them?

**Strengths And Weaknesses:**

### Strengths

* The paper is easy to read and self-contained. Anyone with reasonable in background VLM's can follow it without having a background in adversarial attacks.
* The number of experiments are extensive and hence is quite convincing. It seems the authors have put a lot of effort in testing their hypothesis.
* This paper will have a audience on researchers working on data poisoning + VLM's.


### Weaknesses

Given that the paper is appropriately scoped, I don't think there are any major weaknesses and is mostly good. However, I have a few requested critical changes and changes that can strengthen the paper.

---

> ### Author Response · Authors · 2024-10-14
> **Addressing the Changes Requested by the Reviewer**
>
> We thank the reviewer for the time and effort to review our work. We are glad that the reviewer found our paper to be self-contained and our experiments to be extensive. We address the limitations here:
>
> > 1. Replace "stronger pretraining objectives" with self-supervised losses in the text
>
> Thank you for your feedback. We will revise the paper to claim what matches our experiments.
>
> > 2. How much were the higher accuracies for the model pre-trained with stronger objective
>
> Among the models that we trained, the one trained with MMCL + SSL was usually about 1% more accurate than the models trained with just MMCL. This would not make a difference in the main figures as the accuracies are still similar. However, for the experiment with ViTs (which was suggested in point 7 below), we cleaned the highest accuracy models for both pre-training losses and the conclusion from the experiments are the same.
>
> > 3. Change the legend for clarity:
>
> Thank you for this suggestion. We will replace the legends throughout the paper for clarity.
>
> > 4. Add intuition about why incorporating SSL during pretraining renders CleanCLIP ineffective. My understanding is as follows. "CleanCLIP focuses on breaking correlation between by having meaningful unimodal representations during fine-tuning. Since the SSL loss is also applies on the poisoned images, during pretraining, the unimodal representations can also be implicitly associated with the target image."
>
> We thank the reviewer for suggesting this and we agree with their intuition. We will add this in the paper.
>
> > 5. I assume the paper borrows the augmentation hyperparameters from CleanCLIP. What happens if stronger augmentation like RandAugment is used for pre-training?
>
> We indeed borrowed the augmentation hyperparameters from CleanCLIP. We performed experiments with RandAugment in SSL. Concretely we pre-trained three CLIP models from scratch with increasing augmentation magnitude: 10, 20, and 30 (out of a total of 31 magnitude buckets). We trained the models for 64 epochs and plot the Imagenet validation zeroshot top-1 accuracy and ASR in the figures here: https://imgur.com/a/vUpqHK9.
> We note the ASR in these graphs are very noisy with large jumps in values, whereas the accuracy is still smooth, meaning that stronger augmentation did have some effect on ASR, but not monotonic.
>
> > 6. What happens if the self-supervised loss during pretraining is applied only on clean images? One can imagine a scenario where the contrastive model is trained on uncurated data while practitioners can apply the SSL loss on a small set of curated images (For example, ImageNet)
>
> Thank you for suggesting this experiment. We believe this scenario will not lead to higher accuracies compared to the case when the same model trained with self-supervised loss on the large uncurated dataset and due to our limited academic resources, we have used that to conduct experiments for other weaknesses (below).
>
> > 7. The experiments can also be strengthened by having one experiment on ViT models
>
> Thank you for suggesting this experiment. We trained two ViT models, one with MMCL and another with MMCL + SSL (on CC6M dataset with 3000 poisoned datapoints), and cleaned them using CleanCLIP. Here is the cleaning plot for these models: https://imgur.com/a/o6lUGY2.
>
> We observe that in this case both the MMCL and MMCL + SSL pre-trained models experience a significant drop in their accuracies in order to get a low ASR. The accuracy of the model pre-trained with MMCL drops by 30% (relative) and the accuracy of the model pre-trained with MMCL + SSL drops by 41% (relative).
>
> > 8. What is the precise difference between BadNet and "label-consistent" data poisoning?
>
> In a BadNet attack, the backdoor trigger is added to any image and the captions of those corresponding images are modified to the label “banana”. In case of label-consistent poisoning we only add the backdoor trigger to the images whose captions already mentioned banana (exactly as you mentioned).

---

### Review · Reviewer_Noxh · 2024-08-14

**Summary Of Contributions:**

In this paper, the authors explore the effectiveness of the previously proposed CleanCLIP defense to remove the effect of poisoned images from a pretrained multimodal model when trained with a stronger objective of combining MMCL and SSL loss, instead of simply MMCL loss.

Through extensive experimentations, they show that CleanCLIP is effective when used on a model that is pre-trained with MMCL loss, while it performance significantly worse when used on model pre-trained with MMCL and SSL loss. They show this effect replicates consistently through different setups, including a) different datasets, b) different ways poison can be induced, c) stopping criterion during cleaning, d) proportion of poisoned images in cleaning dataset, and e) impact of strategies like additional regularization or stronger loss function during cleaning.

The work makes an important point on the challenge in creating a robust defense to remove backdoors from pretrained models.

**Audience:**

Yes

**Broader Impact Concerns:**

None.

**Claims And Evidence:**

Yes

**Requested Changes:**

1. In section 5, while evaluating an upper bound on poison removal, instead of using pseudo-labels why not use the true labels? Or use labels as predicted by a really good image classification model in case true labels are not available.
2. Can you clarify why the models are pre-trained for 64 epochs? If possible add experiments when either model is pre-trained for fewer epochs or with even smaller number of poisoned samples in the dataset.
3. Can you maybe explore and compare the structure of the embeddings or the activations in last couple of layers in the model when an image with the trigger and without trigger are passed for both MMCL and MMCL+SSL pretrained models. I hope to see if it provides any intuition into why CleanCLIP fails in the later case.
4. Lastly, can you mention if the code will be released upon publication? If open-sourced it will also be an important contribution to the field and community.

**Strengths And Weaknesses:**

Strengths

1. Well-written paper with clearly stated objectives, attack model and contributions.
2. Extensive experimentation in applying CleanCLIP with different setups, with hyperparameter search for learning rate in these setups. Transparency in the experimentation setup, including all parameters used is appreciated.
3. Experimentation to improve CleanCLIP through stronger objectives in fine-tuning than pre-training, showing that even these do not work when model is pre-trained with MMCL + SSL loss.

Weaknesses

1. In section 5, the subsection on *Cleaning using an Objective distinct from Pre-training* is hard to understand on how exactly pseudo-labels and created. And it is not clear why are we using DeepClust as an additional loss instead of directly using true labels, or labelling the images using a much stronger Classification model and using that for training.
2. A single pre-trained model setup is used throughout the experimentations to clean the model (though with different datasets CC3M and CC6M). The number of epochs used in the setup seems extremely high compared to epochs used in practice.
3. It is still not very clear on why the CleanCLIP works with MMCL pre-trained model but not when pre-trained with MMCL+SSL loss.

---

> ### Author Response · Authors · 2024-10-14
> **Addressing the Changes Requested by the Reviewer**
>
> We thank the reviewer for their time and effort in reviewing our work. We are elated to know that the reviewer found our paper to be well-written, the experiments to be extensive with multiple hyperparameters and transparent in design. We address the weaknesses here:
>
> > In the subsection on Cleaning using an Objective distinct from Pre-training is hard while evaluating an upper bound on poison removal, instead of using pseudo-labels, why not use the true labels or labels as predicted by a really good image classification model?
>
> Thank you for pointing this out. In this particular subsection we used two kinds of labeling schemes for the DeepClustering loss. One was the actual deep clustering loss using the vision encoder of the model, and another was a cheating experiment to establish the upper bound. For the cheating experiment we indeed use labels as predicted by a really good image classification model. In the second paragraph in this subsection we mention using SigLIP ViT-L/14 for this task which has a zero-shot Imagenet accuracy of 83.08%. We used this model as it had the highest image classification accuracy of any CLIP style model as per OpenCLIP repository: (https://github.com/mlfoundations/open_clip/blob/main/docs/openclip_results.csv). We do have access to the true labels for these images as they are collected from the internet.
>
> > 2. A single pre-trained model setup is used throughout the experimentations.
>
> We agree that we use a single pre-trained model for the experiments; however, we show the same phenomenon with several different settings like datasets, pre-trained from scratch vs. pre-trained with finetuning). Our experimentation design closely resembles CleanCLIP’s experimental design to facilitate a fair and accurate comparison.
>
> > 3. The number of epochs used in the setup seems extremely high compared to epochs used in practice. Can you clarify why the models are pre-trained for 64 epochs?
>
> Thank you for pointing this out. We followed the exact same setup as used in the CleanCLIP paper for a fair and accurate comparison, and therefore we pre-trained the models for 64 epochs.
>
> > 4. add experiments when model is pre-trained for fewer epochs or even smaller number of poisoned samples in the dataset
>
> Thank you for suggesting this experiment. We trained two models with 800 poisoned datapoints with the CC6M dataset (for context in the paper, we trained models with 3000 poisoned datapoints). We trained the models for a lower number of epochs, 38 epochs. We then cleaned them using CleanCLIP. Here is the plot of the cleaning results: https://imgur.com/a/9Qhn1Ys. We note that irrespective of the number of poisons and epochs, we observe the same phenomenon as reported in the paper.
>
> > 5. It is still not very clear on why the CleanCLIP does not work with MMCL + SSL loss. Can you maybe explore and compare the structure of the embeddings or the activations in the last couple of layers in the model when an image with the trigger and without trigger are passed for both MMCL and MMCL+SSL pretrained models.
>
> We plot the UMAP dimensions of the embeddings of 24000 images with and without the trigger using two models, one poisoned with MMCL and another with MMCL + SSL: https://imgur.com/a/8It30tf. We see that in both the cases, the embeddings of the two sets occupy distinct parts of the space. However, just these UMAP plots do not provide additional intuition about the difference in performance in the two pretraining losses.
>
> > 6. Can you mention if the code will be released upon publication?
>
> Thank you for asking! We will absolutely release all the code and the data upon publication of the paper.

---

> > ### Comment · Reviewer_Noxh · 2024-11-22
> > **Quick Feedback**
> >
> > I thank the authors for their response. Just a minor point, given that authors have access to ground truth labels, for cheating experiment it does make sense to use these to show the limit of what is possible if the model used for cheating was perfect.

---

### Review · Reviewer_uaEK · 2024-10-04

**Summary Of Contributions:**

The paper studies the effectiveness of an existing defenses (CleanCLIP) for removing backdoors in contrastively trained image-text models (CLIP-type). The main contributions of the paper are in providing a better understanding of when (and when not) CleanCLIP is effective in removing backdoors from a pretrained backdoored CLIP model. In particular, the authors show that the pretaining objective has a major effect on the downstream effectiveness of CleanCLIP: while CleanCLIP is effective for a depoisoning a purely contrastively trained CLIP model, it is far less effective when pretaining also includes a self-supervised loss on image/text augmentations. Moreover, they study settings of practical interest such as removing backdoors without assuming a "golden" backdoor-free fine-tuning dataset. The authors focus on CC3M and CC6M for pretraining of CLIP and ImageNet-1K zero-shot for evaluating attack-success rate.

**Audience:**

Yes

**Broader Impact Concerns:**

Removing backdoors from a pretained model is generally considered desirable; however, there might also be certain cases where a model provider has a legitimate reason for adding a backdoor to a model (e.g. for being able to check if a third party uses their particular model outside of the model's usage restrictions)---in this case, effective backdoor removal might be considered less desirable. I don't think this is overly problematic, but the authors could briefly discuss this topic as part of a broader impact discussion.

**Claims And Evidence:**

Yes

**Requested Changes:**

Please address the weaknesses listed above.

Required for acceptance:
- Tone down claims or study additional defenses
- Evaluate on more tasks than just ImageNet-1k zero-shot
- Improve accessibility of scatter plots (in particular Figure 2, 5, and 7)

Nice to have:
- Add more structure to Section 4 and 5 by adding subsections
- If feasible, provide a better understanding of why CleanCLIP is less effective on models trained with a self-supervised loss.

**Strengths And Weaknesses:**

Strengths:
 + Defending against backdoors in models trained on un-curated web data is a practically relevant problem
 + The authors conduct a systematic study that sheds light on when and when not the state-of-the-art defense CleanCLIP is effective in removing backdoors. The authors provide reasonable ablations and control experiments on their main findings in Section 5.
 + The authors additionally study a realistic setting in which even the fine-tuning dataset contains some backdoor examples as well as a setting where ASR cannot be evaluated during cleaning. These settings are of higher importance for practitioners
 + The paper is well-written and easy to follow

Weaknesses:
 - Focussing only on a single defense (CleanCLIP), even if this is the state-of-the-art defense, is very limiting. In particular, it does not justify broader claims such as, e.g.,  the title "Effective Backdoor Mitigation in Vision-Language Models Depends on the Pre-training Objective". I would encourage authors to either tone down claims or study additional defenses.
 - The paper lacks a good hypotheses of why a backdoored model that was trained with an additional self-supervised loss term is substantially harder to clean. This could for instance be because of the image augmentations being also applied to the backdoor-trigger (and thus making the model more robust in detecting the trigger itself). A useful control experiment would be to add the trigger always after image augmentations (instead of before).
 - Evaluating poisoned and cleaned models on only a single downstream task such as zero-shot classification on ImageNet-1k is too limited. Experiments on additional tasks would strengthen the main claims of the paper.
 - Section 4 and 5 are rather lengthy without any substructure. This makes it unnecessarily difficult to navigate the paper.
 - Figure 2, 5, and 7 are not accessible because of too similar colors/too small size of the dots in the scatterplot.

---

> ### Author Response · Authors · 2024-10-14
> **Addressing the Changes Requested by the Reviewer**
>
> We thank the reviewer for their time and effort in reviewing our work. We are glad to know that the reviewer found our paper to be well-written and easy to follow. We address the weaknesses here:
>
> > 1. Tone down claims or study additional defenses
>
> Among the set of works that propose a defense against multimodal backdoors:
> Robust Contrastive Language-Image Pretraining against Data Poisoning and Backdoor Attacks
> Semantic Shield: Defending Vision-Language Models Against Backdooring and Poisoning via Fine-grained Knowledge Alignment
> Better Safe than Sorry: Pre-training CLIP against Targeted Data Poisoning and Backdoor Attacks
> CleanCLIP: Mitigating Data Poisoning Attacks in Multimodal Contrastive Learning
>
> Only CleanCLIP is a post-hoc defense strategy, i.e., it is the only proposed technique that given a poisoned model can clean it via finetuning. All other defense strategies are train-time strategies that can only be applied when training a model from scratch. And therefore we only used CleanCLIP as the method to clean our poisoned model. We agree with the reviewer’s concern regarding the limitation and are happy to narrow down the claim only to be CleanCLIP specific. We would do this in the abstract and leave the title unchanged to encourage future works to consider this case when proposing novel defenses.
>
> > 2. Evaluate on more tasks than just ImageNet-1k zero-shot
>
> We evaluated only on ImageNet-1K zero-shot because that was the only task CleanCLIP evaluated on, and therefore it was the only fair comparison we could make. As per reviewer’s request, we also evaluated the ASR and accuracy on two more datasets: Caltech101 and Cifar-100. Here are the plots: https://imgur.com/a/NCxICVU. We observe a similar phenomenon with models trained with MMCL and MMCL + SSL and evaluated on Caltech101 and Cifar-100 datasets as in the case of ImageNet-1K.
>
> > 3. Improve accessibility of scatter plots (in particular Figure 2, 5, and 7)
>
> Thank you for pointing this out, we will update the plots in Figure 2, 5, 7, and 8 to have a larger marker size, this will improve the accessibility of the plots.
>
> > 4. Add more structure to Section 4 and 5 by adding subsections
>
> We used bolded paragraph names: “Cleaning using an Objective distinct from Pre-training” to structure sections 4 and 5, we would change them to subsections for better readability.
>
> > 5. If feasible, provide a better understanding of why CleanCLIP is less effective on models trained with a self-supervised loss.
>
> Our hypothesis of why CleanCLIP fails to work on stronger pre-training objectives align with your intuition about image augmentations being applied to the backdoor trigger and thereby ingraining the poison more strongly (and this also aligns with Reviewer qLae’s intuitions). While it will be interesting to conduct the experiment the reviewer suggested, given our limited academic resources, we have used that to conduct the experiment which were required for acceptance (Point 2).

---

> > ### Comment · Reviewer_uaEK · 2024-10-18
> > **quick feedback**
> >
> > I would like to thanks the authors for their feedback, which has addressed my concerns. Given that the urgent points have been addressed, it would still be valuable to run an experiment on point 5 (better understanding) now - it would certainly strengthen the paper (as said, it is not required imo for acceptance, so I leave it to the authors but encourage conducting this experiment)

---

### Author Response · Authors · 2024-10-14
**Summary of Author Response**

Dear Reviewers and Action Editor,

We sincerely thank you for the time and effort you put in to review our work so thoroughly! We are glad to find that all reviewers agreed that our paper to be well-written, easy to follow, and had extensive experimentations. We have incorporated all suggestions including writing changes, figure readability, and further strengthening of the experiments (to the best of our limited academic ability).

All of these details will be updated in the manuscript and the code will be released upon publication.

We look forward to hearing from your,

Sincerely,
#2933 Authors

---

### Decision · Action_Editor_kWwM · 2024-11-25

**Recommendation:** Accept with minor revision

**Comment:**

Reviewers are overall very interested in the results presented in the paper. After revision, most of the concerns are addressed, although they have some small editorial points remaining to be addressed, and I recommend the authors to revise the paper accordingly re reviewers' final recommendations as detailed below:

(i) "While there is a small recommendation that would improve the quality of work, which is for the experiment where they assume they have the image labels. In that experiment, they could use ground truth instead of another model's output. If we want to cheat, better go all the way to show what happens in the extreme limits."

(ii) "For the final version, the authors can highlight the fact that CleanCLIP itself seems to fail on VIT models, which is quite interesting. They can also address the promised minor changes in the revision (Replacing terms like "stronger" with more precise "SSL losses" and cleaning up Figure 3.)"

Some reviewers are wondering whether the scope of the paper (studying CleanCLIP only) is a bit too narrow. Quote Reviewer uaEK "Focussing only on a single defense (CleanCLIP), even if this is the state-of-the-art defense, is very limiting". As per TMLR submission guideline, I personally think that a narrower scope of study is fine as long as the presented evidence is solid, for which reviewers agree that this paper is good regarding this aspect. However to better inform the wider CV/ML community (rather than security+AI), I recommend the authors to (1) make their claims regarding CleanCLIP precise and clear, and (2) discuss potentially whether the findings regarding CleanCLIP can be extended to other type of backdoor mitigation methods sharing similar ideas (and precisely which ideas) of CleanCLIP.

Also on formatting issues: now the abstract part of the paper doesn't seem to follow the abstract format. Please revise.

**Audience:**

Computer vision and machine learning researchers caring the robustness properties of their models.

**Claims And Evidence:**

The paper studies the effectiveness of CleanCLIP as a defence method for backdoor attacks to contrastively trained image-text models (CLIP-type), showing when (and when not) this approach is effective. The main claim is that the pre-training objective for CLIP-style models has a major effect on the downstream effectiveness of CleanCLIP, and the claim is supported by solid experiments a number of backdoor attacks and a few SSL pre-training objectives.